# Dual-Energy Processing of X-ray Images of Beryl in Muscovite Obtained Using Pulsed X-ray Sources

**DOI:** 10.3390/s23094393

**Published:** 2023-04-29

**Authors:** Alexander Komarskiy, Sergey Korzhenevskiy, Andrey Ponomarev, Alexander Chepusov

**Affiliations:** Institute of Electrophysics, Ural Branch, Russian Academy of Sciences, Yekaterinburg 620016, Russia; sk@iep.uran.ru (S.K.); avponomarev@ya.ru (A.P.); chepusov@iep.uran.ru (A.C.)

**Keywords:** dual-energy X-ray imaging, X-ray absorption, pulsed X-ray source, computed tomography, beryl, muscovite, image processing

## Abstract

This paper presents the development of a method for dual-energy processing of X-ray images using pulsed X-ray sources for the contrast detection of beryl in muscovite mica in 2D X-ray and CT images. These substances have similar chemical properties and are difficult to differentiate when one is against the background of the other using methods based on X-ray absorption. In the experiments, we used three pulsed X-ray sources with different maximum voltages. We performed modeling of the emission spectra and selection of the necessary energy bands due to X-ray absorbing filters: a positive effect was shown for dual-energy image processing when the function of converting X-ray radiation into a signal using the VIVIX-V 2323D detector was taken into account. As a result, a pulsed X-ray source with the pulse voltage of 330 kV was chosen for the contrast detection of beryl, with the content of 5–7% against the background of muscovite and the thickness up to 70 mm. Using this source and the developed mathematical algorithms, it is possible to obtain a band of low-energy radiation at the level of 70–80 keV, as well as high-energy radiation in the range of 180 keV. Methods based on the X-ray absorption can become both additional and independent methods for studying and monitoring membranes; these objects range from tens of nanometers to several micrometers in size.

## 1. Introduction

X-ray fluorescence (XRF) is a well-established method used in the mining industry to separate valuable minerals. It has proven successful in prospecting for minerals such as diamonds and emeralds, which emit detectable optical spectrum radiation when they are exposed to X-rays [1,2]. Optical radiation is absorbed in the waste rock, which is why one can only detect fluorescence coming from minerals which are not covered in rock. In order to find hidden gems and minerals, it is necessary to sequentially crush the ore up to the point when the target surfaces. Crushing often causes gemstone damage, resulting in a reduction in their size and the lose of some of their value.

Some research papers suggest using X-ray absorption methods in the process of searching for valuable minerals [3,4]. These methods are based on the differences in X-ray absorption characteristics of different substances [5]. The optimal results are achieved when studying complex substances with widely varying X-ray absorption properties [6]. These methods provide information on the internal structure and composition of the research object. The ore and the mineral it contains tend to have a very similar composition of chemical elements and density, and, therefore, demonstrate similar X-ray absorption properties, making the contrast between the target and the waste rock barely noticeable in 2D projection images. This principle applies to diamonds in kimberlite and emeralds (beryl) in mica.

The current level of technology supports further development of the dual-energy absorption-based X-ray imaging method. Dual-energy image processing uses the energy dependence of X-ray attenuation, which provides more information about materials than standard X-ray radiography methods.

The method is based on the difference which occurs when substances absorb X-rays of different energies. The absorption of low- and high-energy X-rays of substances with different atomic numbers changes disproportionately. At present, dual-energy X-ray image processing is widely used in healthcare and security inspections, and has promising applications in non-destructive testing and the mining industry [7,8,9].

Dual-energy processing of projection images allows us to obtain 2D images suitable for detecting a target with a certain atomic composition. However, this method has limitations related to the dimensions of targets and differences in their X-ray absorption characteristics. The entire internal structure may be captured using the computed tomography method [10]. This method was initially developed to be used in healthcare, but it is now widely applied in security inspections, as well as in other areas, such as geosciences and mining [11,12,13]. The dual-energy method is also used extensively in areas related to multiphase flow meters [14]. Computed tomography (CT) is a non-destructive testing method that enables three-dimensional visualization of the internal structure of a target (e.g., a rock sample). Several companies offer CT analysis services for mining enterprises designed, for example, to detect gold in hidden tectonic structures [15].

Dual-energy methods are also used in computed tomography to detect the research object and identify its exact shape. The best possible case-specific radiation energy values must be selected for each study [16]. In rocks, computed tomography allows us to build a shape model of the target against the background of waste rock to facilitate its subsequent extraction without damage. CT also has certain inherent limitations due to the thickness and shape of the target and atomic composition of the substances to be detected. For example, a number of studies carry out dual-energy separation of diamonds hidden in kimberlite using model examples with a number of simplifications [17]. Therein, the dual-energy method is used to separate fine kimberlite from the target diamond; the kimberlite powder and the diamond are placed into a cylindrical container, i.e., symmetrical geometry is in place. The diamond consists only of carbon, which differs significantly in its atomic number from kimberlite.

In the current study, experiments were conducted to separate beryl from mica using the dual-energy methods of imaging processing for 2D X-ray projections and 3D CT images. In terms of its chemical composition, beryl is almost indistinguishable from emeralds.

When one obtains images using different spectral distributions, it is critical to correctly form the required energy spectra, taking into account the objects of study and their thickness [18]. The main difficulty is that X-ray radiation has a continuous spectrum, while deviations from monoenergetic radiation negatively affect dual-energy processing of X-ray images. The existing theoretical studies of the dual-energy image processing methods presuppose that the radiation is close to monoenergetic.

Our laboratory is developing pulsed X-ray sources with a high pulse repetition rate based entirely on solid-state electronic components [19,20,21]. Instead of a gas-filled spark gap, the last circuit of our high-voltage pulse generator has a semiconductor opening switch (SOS) [22,23,24]. This allows us to generate X-ray pulses with a high repetition rate and sets our generator apart from other pulsed X-ray radiation sources. Pulsed X-ray sources, including those presented in this paper, have high peak power and short flash duration (tens of nanoseconds) and use two-electrode cold cathode X-ray tubes. The fraction of low-energy quanta in the spectra of these pulsed sources is higher compared with the bremsstrahlung spectra of X-ray sources with a filament cathode [25]. Due to the short radiation flash and a high pulse repetition rate, these sources may be used for capturing 2D projections of objects in motion [26], for example, on a conveyor belt. This paper presents methods for separating energy bands for further use in dual-energy transformations of X-ray images.

It is worth mentioning that certain existing methods allow us to analyze several regions of the spectra (the so-called color or spectral visualization) [27]. These methods of study are developing using detectors that are capable of registering different regions of the spectra simultaneously. This technology, however, is based on photon counting [28], which is not applicable for rapid processes and may not be used with pulsed radiation sources. In addition, such detectors register low-energy X-rays radiation and are not suitable for studying rock with the thickness of tens of millimeters.

These studies make it possible to select the optimal energy band of radiation for the contrast-enhanced detection of beryl against the background of mica. Sources with different peak voltages and power values have been used within this research. Automating these studies will allow us to optimize the process of searching for precious minerals hidden in the host rock and subsequently facilitate the recovery of larger gems. These studies will provide valuable insights into the fundamental processes behind the formation of gem minerals in host rock. This research contributes to developing methods of dual-energy X-ray image processing, both in terms of hardware (the radiation source and the detector) and suggesting a method for separating specific energy bands from the emission spectra of a pulsed X-ray source.

In a number of existing studies, dual-energy processing of X-ray images is used to detect substances with dimensions starting from tens of nanometers [29,30]. The devices and methods based on dual-energy X-ray image processing and presented herein may be further applied to study and analyze thin films and coatings. Upon further development, these methods, which focus on the absorption of X-rays by substances, may serve as both supplementary and independent methods for studying and monitoring objects with dimensions ranging from tens of nanometers to several micrometers.

## 2. Materials and Methods

### 2.1. Beryl and Muscovite

This study used beryl and muscovite mica mined in the Urals. The chemical formula of beryl is Al_2_[Be_3_(Si_6_O_18_)]. It contains 14% beryllium oxide, 19% aluminum oxide, and 67% silicon dioxide. In the presence of low concentrations of certain impurities, beryl becomes an emerald, i.e., a precious gem. In nature, beryl is found in intergrowths with muscovite mica.

Mica is a frequent companion of mineral rocks in the Earth’s crust and was formed from cooled molten lava. In its natural form, it occurs as white or light gray units that may split into fine hard plates and scales with high light reflectivity. The chemical composition of muscovite is KAl_2_[AlSi_3_O_10_](OH)_2_. Muscovite mica is a material containing a high percentage of aluminum oxide (up to 39%) and silicon dioxide (over 45%) and over twenty impurities made up of other chemical elements and compounds. Figure 1 shows the intergrowth of beryl in muscovite mica used in some of the experiments.

To calculate the attenuation coefficient, we used only the elements from the given chemical compositions, since the research objects have a small amount of impurities that do not significantly affect the changes in absorption properties. These substances contain a substantial proportion of aluminum, oxygen, and silicon. The densities of beryl and muscovite mica were similar and amounted to 2.7 and 2.9 g/cm^3^, respectively. The difference in the mass X-ray attenuation coefficients between beryl and muscovite was explained using a higher concentration of aluminum Al and silicon Si in muscovite mica, as well as through the presence of beryllium (Be) in beryl, which had a very low X-ray absorption coefficient.

### 2.2. Dual-Energy Image Processing

We considered a case with monoenergetic X-ray radiation [31]. Attenuation of monoenergetic radiation passing through a homogeneous object is described as follows:(1)$$I=I_{0}\cdot e^{\left(-µ*\rho *x\right)},$$
where *I* is the intensity of radiation exiting the object; *I*_0_ is the direct radiation intensity; *µ* is the mass attenuation coefficient in a given substance for a given energy; *ρ* is the density, which may be assumed as a constant for a particular substance; and *x* is the target thickness. If the target consists of an *n* number of materials, then Expression (1) will take the following form:(2)$$I=I_{0}\cdot e^{\overset{n}{\underset{i=1}{\sum }}(-{µ}_{i}\cdot \rho_{i}\cdot x_{i})},$$

If the research object consists of two superimposed substances and the thickness is not known, two X-ray sources shall be used (a low-energy source *E_L_* and a high-energy source *E_H_*). In the case of the superposition of two different objects and use of two monoenergetic radiation sources, the attenuation of radiation may be described using the following system of equations:(3)$$\left\{\begin{array}{c} I_{L}=I_{L\_0}\cdot e^{\left(-\mu_{L\_1}\cdot \rho_{1}\cdot x_{1}-\mu_{L\_2}\cdot \rho_{2}\cdot x_{2}\right)}\\ I_{H}=I_{H\_0}\cdot e^{\left(-\mu_{H\_1}\cdot \rho_{1}\cdot x_{1}-\mu_{H\_2}\cdot \rho_{2}\cdot x_{2}\right)} \end{array}\right.$$

Indices 1 and 2 refer to the two different substances (beryl and muscovite mica, respectively), while indices *L* and *H* correspond to the values of the variables at low and high energy, respectively. Upon a substitution of *m* = −ln(*I*/*I*_0_), the system of Equation (3) takes the following form:(4)$$\left\{\begin{array}{c} m_{L}=\mu_{L\_1}\cdot \rho_{1}\cdot x_{1}+\mu_{L\_2}\cdot \rho_{2}\cdot x_{2}\\ \;m_{H}=\mu_{H\_1}\cdot \rho_{1}\cdot x_{1}+\mu_{H\_2}\cdot \rho_{2}\cdot x_{2} \end{array}\right.$$

The mass attenuation coefficients depending on the radiation energy for beryl (Al_2_[Be_3_(Si_6_O_18_)]) and muscovite mica (KAl_2_[AlSi_3_O_10_](OH)_2_) are shown in Figure 2a. In actual practice, it was very difficult to distinguish between these substances in an X-ray image. For effective separation, the substances required different X-ray attenuation coefficients. Significant differences in mass attenuation coefficient *µ* were only observed for low-energy radiation; however, radiation with energies below 40 keV was largely absorbed by these substances at thicknesses of several millimeters or more. This fact is clearly demonstrated via the dependence curve for the intensity of radiation exiting the sample and the radiation intensity of the direct unattenuated beam as a function of energy. This dependence for the same thickness of beryl and muscovite, which equals 10 mm, is shown in Figure 2b.

The curves demonstrated more than a ten-fold attenuation for radiation energies below 40 keV due to absorption in the samples. It was almost impossible to obtain X-ray images with a high signal-to-noise ratio at such low energies, even at a very high radiation dose. Furthermore, in actual practice, large thicknesses of these minerals up to 70 mm had to be examined. A slight bend was observed on the curve of absorption intensity for the energy range of approximately 90 keV, after which the absorption intensity reduced, becoming less pronounced as compared with the 50 keV range. At low energies, a change in energy of several kilo-electronvolts led to a several-fold change in the absorption coefficient.

A very rapid change in the attenuation coefficient at radiation energies up to 40 keV complicated the imaging process even for objects with a thickness of several millimeters. Sources of monoenergetic radiation have limited application due to a number of factors, one of which is their low radiation intensity. Synchrotrons, with high radiation power and spectra similar to a monoenergetic one, are mainly applied for solving scientific problems and are not easily available. In practice, X-ray sources with a bremsstrahlung spectra are used, and the spectral distribution is adjusted using X-ray absorbing filters and voltage adjustments applied to the X-ray tube. Such radiation, in turn, differs from monoenergetic radiation: low-energy radiation is attenuated more strongly than high-energy radiation when it passes through the object.

For a single substance and monoenergetic radiation, system of Equation (4) takes the following form:(5)$$\left\{\begin{array}{c} m_{L\_1}=\mu_{L\_1}\cdot \rho_{1}\cdot x_{1}\\ \;m_{H\_1}=\mu_{H\_1}\cdot \rho_{1}\cdot x_{1} \end{array}\right.$$

It follows that
(6)$$m_{H\_1}=\frac{\mu_{H\_1}}{\mu_{L\_1}}\cdot m_{L\_1}$$

It is obvious that the dependence of *m_H__*_1_(*m_L__*_1_) is a straight line with the slope factor *µ_H_*__1_/*µ_L_*__1_. The changes in *m* occurred due to variations in the thickness of the object. We plotted these dependencies for the substances under study (beryl and muscovite mica) through selecting different values pairs of low *E_L_* and high *E_H_* energy. The object thicknesses were in a range between 1 and 70 mm. Figure 3 shows the dependences for the following pairs of energy values: 65 and 180 keV, 85 and 180 keV, 125 and 180 keV, and 65 and 85 keV.

The larger the angle between the curves, the higher the probability of detecting the target that was interspersed in another object (in our case, the probability of detecting embedded beryl in muscovite). The figure shows that the angle between the curves is small when there is a slight difference between the low and high energy values. This finding can be seen for the following pairs of low and high energy, respectively: *E_L_* = 125 keV and *E_H_* = 180 keV; *E_L_* = 65 keV; and *E_H_* = 85 keV, i.e., the absorption of radiation in beryl and muscovite for these energy values varies proportionally.

If the target was partially composed of beryl and muscovite mica, the point was located between the curves corresponding to the selected energies. Respective points (diamond symbols) are shown in Figure 3 and refer to the case when the thickness of beryl is 7 mm and the thickness of muscovite is 10, 15, and 25 mm. These points were calculated at the low energy of 85 keV and the high energy of 180 keV. The points lay between the curves; the dotted lines showed the object thicknesses.

### 2.3. Dual-Energy Image Processing with Pulsed X-ray Source

The calculations for monoenergetic sources were close to the ideal case and illustrative of the differences between beryl and muscovite visible in X-ray images, depending on the object thickness and radiation energies. In actual practice, sources with a bremsstrahlung spectrum were used. A selection process was, therefore, described for the radiation energy bands of low and high energies. In the figures below, one can see oscillograms of current and voltage pulses (Figure 4a) and the calculated spectra of a pulsed X-ray source with the maximum voltage of 330 kV (Figure 4b) used in the study. The spectra were calculated via differentiation using the current and voltage oscillograms. The total spectra were the sum of the spectra obtained for the current and voltage at each point in time (depending on the acquisition time of the oscillograph; in this case, it was 0.1 ns). The characteristic radiation was not considered, since, according to our estimates, the intensity of the characteristic lines accounted for less than 1% of the entire spectra intensity.

In the bremsstrahlung spectra, there was a high proportion of low-energy quanta, the absorption coefficient *µ* depended on the energy, and Expression (1) took the following form:(7)$$I=\overset{\lambda_{0}}{\underset{\lambda =\lambda_{max}}{\sum }}I_{0\_\lambda }\cdot e^{-\left(\mu_{\lambda }\cdot \rho \cdot x\right)}$$

The total intensity of radiation exiting the substance was the sum of all intensities in the spectra, considering the attenuation for a specific wavelength, or more correctly, for a small variation interval of the wavelength.

In actual practice, the energy band separation for X-ray sources was performed through changing the source voltage and/or using X-ray absorbing filters that cut off the low-energy quanta. The radiation in these cases had a pronounced energy band. With a pulsed X-ray source, it became difficult to change the voltage due to the use of a two-electrode X-ray tube with a cold cathode. In case of low energy, we used the calculated spectra, obtained via subtracting the fraction of radiation exiting the copper filter from the total spectra with a correction factor, which allowed us to account for the reduction in the radiation intensity due to absorption through the filter [18]. However, for the sake of simplicity, radiation exiting a 1–3 mm aluminum filter was used as it cut off a significant fraction of low-energy quanta. We developed mathematical algorithms that remove the high-energy radiation band from this spectrum. This result was achieved due to passing through a copper filter. The spectra defined after using the filter and applying the high-energy radiation removal algorithm are shown in Figure 5. The emission spectra of direct current sources were similar to the spectra of pulsed sources, with the difference being that, while accelerating voltage of 330 kV, the peak of emission intensity for the continuous source shifted to the high-energy range. Therefore, with continuous sources of radiation, the entire spectrum was less applicable for the low energy.

Considering the dependence between the mass attenuation coefficient and energy, the system of Equation (3) takes the following form:(8)$$\left\{\begin{array}{c} I_{L\_p}=\overset{\lambda_{0\_L}}{\underset{\lambda =\lambda_{L\_max}}{\sum }}I_{0L\_\lambda }\cdot e^{\left(-\mu_{1\_\lambda }\cdot \rho_{1}\cdot x_{1}-\mu_{2\_\lambda }\cdot \rho_{2}\cdot x_{2}\right)}\\ I_{H\_p}=\overset{\lambda_{0\_H}}{\underset{\lambda =\lambda_{H\_max}}{\sum }}I_{0H\_\lambda }\cdot e^{\left(-\mu_{1\_\lambda }\cdot \rho_{1}\cdot x_{1}-\mu_{2\_\lambda }\cdot \rho_{2}\cdot x_{2}\right)} \end{array}\right.$$

Index *p* indicates that the parameter refers to a pulsed radiation source. Numerically, through dividing the spectrum into dλ intervals, we calculated the total intensity for the low- (*I*_0*L_p*_) and high-energy spectra (*I*_0*H_p*_) as the sum of intensities over the entire range of wavelengths. Using these energy dependences, the attenuation in beryl and muscovite at the given thicknesses were then calculated for the high- and low-energy radiation. Figure 6 shows the dependence of m_H_p_ (*m_L_p_*) for the same thicknesses of beryl and muscovite mica that were used in the case of monoenergetic radiation, namely: *m_L_p_* = −ln(*I_L__p*/*I*_0__*p*) and *m_H_p_* = −ln(*I_H__p*/*I*_0_*_p*).

As can be seen, the angle between the two curves corresponding to beryl and muscovite mica is smaller than for monoenergetic radiation. It was more difficult to separate the points where mica of different thicknesses overlapped with the 7 mm beryl from the mica curve than it was in the case of monoenergetic radiation, which is close to the ideal conditions. It may well be expected that, in actual practice, the separation of beryl and muscovite mica was more complicated.

The radiation detector used in the experiments also played an important role. We used a VIVIX-V 2323D flat panel detector with the following specifications: an active area of 120 × 120 mm, pixel size of 185 μm, and scintillator type CsI:Tl. When scintillator substances were exposed to ionizing radiation, they emitted visible light. X-ray radiation was registered as follows: X-ray radiation passed through the scintillator and interacted with its atoms causing them to emit radiation in the optical range of the spectrum, which was efficiently recorded on the semiconductor matrix of the detector. Scintillator light emission efficiency depends on the scintillator type and the X-ray radiation energy. The function *D*(*λ*) for the detector used in this study was calculated on the basis of the applicable experimental data. The relative interaction efficiency is shown in Figure 7a. This curve should be read as follows: the probability of detecting quanta with the energy of 75 keV was 10 times higher than that with the energy of 150 keV, and the probability of detecting quanta with the energy of 50 keV was one and a half times higher than that with the energy of 75 keV. According to the curve, the transformation efficiency strongly depended on the radiation energy.

When we used the function of D(λ), the formula took the following form:(9)$$I_{p}=\overset{\lambda =\lambda_{0}}{\underset{\lambda =\lambda_{max}}{\sum }}I_{0\_\lambda }\cdot D_{\lambda }\cdot e^{-\left(\mu_{\lambda }\cdot \rho \cdot x\right)}$$

With the function of *D*(*λ*), the dependences of *m_L_D_p_* = −ln(*I_L_D_p_*/*I*_0_*D_p*_) and *m_H_D_p_* = −ln(*I_H_D_p_*/*I*_0_*D_p*_) were calculated for beryl and muscovite mica. The dependence curves of *m_H_D_p_*(*m_L_D_p_*) for beryl and muscovite are shown in Figure 7b. As can be seen, the separation efficiency is higher, facilitated via additional energy-based radiation selection at the detector.

## 3. Results

### 3.1. Pulsed X-ray Sources with Different Maximum Voltages

Three experiments were carried out using pulsed X-ray sources to separate beryl against the background of muscovite via the dual-energy methods. The X-ray sources have pulsed cold cathode X-ray tubes. Each experiment uses a separate radiation source. The main difference is that the sources have different maximum voltage values. The first source provides pulsed radiation with quanta energies up to 91 keV [32], the second source provides the maximum radiation with quanta energies up to 330 keV, and the third source provides radiation with quanta energies up to 600 keV.

An intergrowth of beryl with muscovite mica, as shown in Figure 1, is used as the sample. It is placed in a container and covered with sand consisting of 0.1 to 5 mm muscovite particles. The total thickness is 35 mm.

The pulsed X-ray source is located 600 mm away from the VIVIX-V 2323D detector used to register the X-rays. The test sample is placed near the detector. Firstly, the radiation intensity is measured without a filter and the sample. In cases of high-energy radiation, copper filters are used to cut off a fraction of the low-energy radiation. Since the proportion of low-energy radiation is high, unfiltered radiation is taken for the low-energy radiation. The intensity of the radiation exiting the copper filter is then measured. Radiation with this spectral distribution is taken for the high-energy radiation. The processing algorithm then allows the elimination of a fraction of the high-energy quanta from the low-energy spectrum mathematically. The radiation exiting the sample is then recorded under the same conditions (i.e., without a filter for low-energy radiation and with the copper filter for high-energy radiation). The X-ray images obtained with the radiation energies described above are then subjected to dual-energy processing for the contrast-enhanced detection of beryl against the background of muscovite.

The algorithm follows several steps Firstly, a curve is plotted for muscovite based on the experimental data, as shown in Figure 7b for the pair of energies considered, with due consideration of the radiation source. Secondly, separate points on the object are individually compared with the curve on a pixel-by-pixel basis. If the sample contains beryl in addition to muscovite, a deviation from the curve is observed and displayed on the image as a darker pixel (see the enlarged picture in Figure 7b. The processing algorithm is written in Python using open source libraries.

In the first experiment, a pulsed X-ray radiation source with the peak voltage of 91 kV was used, with the pulse current reaching 150 A. Next, a 1.5 mm thick copper filter was used to obtain the high-energy radiation. The images which we created with low- and high-energy radiation are shown in Figure 8a,b. The effective radiation energies have the following characteristics: the low-energy radiation corresponds to monoenergetic radiation of approximately 47 keV, and the filtered high-energy radiation corresponds to the quanta with the energy of approximately 65 keV. Dual-energy processing of these images yields the result shown in Figure 8c.

Beryl in the image is slightly different from muscovite. The thickness of beryl reaches 7–10 mm.

In the next experiment, a pulsed X-ray source with a pulse voltage up to 330 kV was used. The characteristics of this source are shown in Figure 4. The low-energy radiation was obtained using a 3 mm thick aluminum filter. The high-energy radiation was obtained using a 3.5 mm thick copper filter. The previously assumed low-energy radiation has the effective energy of approximately 65 keV (see the low-energy radiation band shown in blue in Figure 5); the radiation exiting the copper filter, with the high-energy quanta, corresponds to the energy of approximately 180 keV. Using a thicker filter strongly reduces the radiation dose and increases the noise. X-ray images for the low- and high-energy radiation are shown in Figure 9a,b, respectively.

The image data was then subjected to dual-energy transformation. The contrast-enhanced detection result for beryl after the transformation is shown in Figure 9c. The contrast is much stronger than for the 91 kV source.

The next experiment was carried out using a pulsed radiation source, with the maximum voltage reaching 600 kV and the current reaching 2 kA. The low-energy radiation was obtained with the help of a 5 mm aluminum filter in order to cut off the quanta with energies below 40 keV. The high-energy radiation was obtained using a 5 mm copper filter. Figure 10 shows the images captured at different radiation energies, as well as the respective dual-energy processing results. The contrast is less pronounced than in the image obtained with the 330 kV source. In terms of effective radiation, the low-energy radiation in this case is equivalent to the energy of approximately 100 keV, and the high-energy radiation is equivalent to the energy over 210 keV. The modeling process for such energies is complicated by the fact that, in actual practice, a significant proportion of scattered radiation emerges, which leads to more noise in the images and a lower contrast.

According to the experimental data, the highest contrast of beryl against the background of muscovite after dual-energy transformation of X-ray images is obtained with the pulsed X-ray source having the maximum voltage of 330 kV.

### 3.2. Experimental Separation of Beryllium against Muscovite of Varying Thickness

We have studied the possibility of detecting beryl with a particle thickness of 1 to 5 mm in muscovite mica with a thickness up to 70 mm. The shape of beryl particles resembles a cube, as shown in Figure 11a. These particles are placed into a mix of 0.1 to 3 mm muscovite particles in a 5 mm thick mold. The thickness of muscovite mica is modified by adding additional mica plates, as shown in Figure 11b. The maximum overall thickness may, therefore, vary between 5 and 70 mm.

A 330 kV pulsed X-ray source was used. The resulting X-ray images of the samples were captured using the distinct radiation spectra obtained after passing a 3 mm thick aluminum filter and a 3.5 mm thick copper filter. Dual-energy transformations were then performed, which aimed to distinguish beryl with maximum contrast. The images shown for the 5 mm mold holding the samples of beryl particles were captured with the radiation after the aluminum (Figure 12a) and copper filters (Figure 12b). Figure 12c shows the resulting image after dual-energy processing.

The result of dual-energy image processing for the embedded beryl particles in muscovite mica up to 70 mm thick is shown in Figure 13. A beryl particle of 2 mm becomes hardly distinguishable from the background as early as at a total thickness of 15 mm. A 3 mm particle loses contrast at a muscovite mica thickness of approximately 25–30 mm. The largest beryl particle of 5 mm remains visible in mica with a thickness up to 70 mm.

The curves in Figure 14 show the ratio of the average brightness of pixels that represent beryl inclusions with thicknesses of 5 and 3 mm to the average brightness of pixels representing muscovite mica, depending on the total thickness.

There is a natural decrease in contrast with increasing mica thickness. The trend lines demonstrate that this decline follows a logarithmic dependence. A 3 mm beryl particle may be visually detected at the mica thickness of 40 mm; however, there are also inclusions with similar contrasts that may be confused with the target substance. A 5 mm beryl particle almost merges with the background at the mica thickness of 70 mm. In a 20 mm piece of mica, beryl particles of 1–2 mm are hardly distinguishable.

### 3.3. Dual-Energy Computed Tomography

More detailed studies of the piece of muscovite with mica inclusions, shown in Figure 1, were conducted using computed tomography. According to the results of the study via the dual-energy method, the highest contrast of beryl detection is observed when using a 330 kV radiation source; therefore, the same source was used for the CT study. Figure 15 outlines the CT experiment scheme. The sample was placed in a cylindrical mold with a diameter of 30 mm and filled with an aqueous solution containing 0.1 to 1 mm muscovite particles, including individual muscovite particles reaching 5 mm. The solution was then dried to obtain a denser structure. The X-ray source generated a fan-shaped divergent radiation beam; the sample was placed on a stepper motor. Two series of images with low and high energies were captured; the energy spectrum was corrected using radiation filters, similar to the experiment that obtained dual-energy images in 2D projections. Reconstruction algorithms from open source Python libraries were used.

The resulting slices for the low-energy radiation in orthogonal projections and 3D are shown in Figure 16. The beryl projection in Figure 16c is similar to the images in Figure 8c, Figure 9c, and Figure 10c, obtained in 2D projections using the dual-energy processing algorithms.

Figure 17a,b show the slices obtained with the two radiation energies. The synthesis of these two images is shown in Figure 17c, where beryl is distinguished against the background of muscovite with a good contrast. In this image, small beryl inclusions are more clearly visible.

## 4. Discussion

This paper provides theoretical calculations aimed at detecting substances with similar chemical compositions (beryl and muscovite mica) using the method of dual-energy processing of X-ray images. We used separation of energy bands from the X-ray spectra to achieve a contrast-enhanced detection of beryl against muscovite background. A pulsed X-ray source with bremsstrahlung spectra was used in the experiments. The low and high radiation energies were separated from the spectra with the help of X-ray filters. When we used a calculated difference in radiation attenuation to distinguish between beryl and muscovite, the distinction was significantly weaker than for the monoenergetic case; the angle between the curves in Figure 6 is very small. In practical application, the radiation detector described above plays an important role. As the calculations have shown, the radiation detector output demonstrates non-linear dependence on the energy of the X-ray quanta. When we take into account the conversion function between X-ray intensity and energy, the distinction between beryl and muscovite increases significantly. This outcome is not at all obvious, as clearly demonstrated through the calculation results presented in Figure 7b.

The experiment with three pulsed radiation sources proved the assumption that, even though the mass attenuation coefficient changes the most at the energies of 10–40 keV, dual-energy processing at a rock thickness of over 10 mm hardly gives any positive results and fails to ensure beryl detection in muscovite. The reason for this finding is that a large proportion of low-energy radiation is absorbed in the sample. As a result, the effective low energy equals approximately 60–70 keV, which is close to the maximum source energy of 91 keV. Attempts to use filters to eliminate the low-energy radiation greatly reduce the radiation intensity and produce a lot of noise at the detector. This case is similar to the theoretical calculations of the dependence m_H_(m_L_) shown in Figure 3 for the monoenergies of 65 and 85 keV, when the angle between the beryl and muscovite curves is extremely small.

Direct experiments for pulsed X-ray sources with different voltages show that the optimal source for dual-energy image processing has a voltage of 330 kV. When we use this source, it is possible to separate the low effective energy of 70–80 keV and the high effective energy in the range of 180 keV. This ensures contrast-enhanced detection of beryl against the background of muscovite in a wide range of thicknesses.

With a 91 kV low-energy radiation source, dual-energy processing results in a weak contrast between beryl and muscovite due to the fact that the radiation is absorbed in the sample. At high energies, when a 600 kV pulsed X-ray source is used, another negative effect occurs: a change in the radiation energy causes absorption in beryl and muscovite to change proportionally.

Radiation with high energies inevitably causes scattered radiation from massive samples. This event reduces the effect of scattered radiation as the research objects are buried in fine sand or immersed in a solution of waste rock sand (muscovite). These measures average the scattered radiation and, in turn, practically eliminate the edge effects. With higher radiation energies, the difference in the absorption of X-ray radiation by the objects under study decreases; the best high effective energy for dual-energy transformations is approximately 180 keV, both for projection images and for CT slices.

The dual-energy method demonstrated good results in beryl detection, against both the background of a solid muscovite piece and a sand mixture of muscovite. Figure 8, Figure 9 and Figure 10 show that the separation process is based on the different compositions of the substances, not on their density, because the density of the sand is much lower than that of the solid piece of muscovite.

The method of dual-energy processing of X-ray images is based on radiation absorption in a substance and enables the assessment of inclusions located within the sample. The experiment aimed at detecting beryl particles of different sizes in mica allows us to establish detection limits through defining the maximum thickness of muscovite for each specific fraction size of beryl. Using the pulsed X-ray source with a maximum voltage of 330 kV, dual-energy X-ray image processing detects 5–7% of beryl in muscovite of up to 70 mm thick.

The CT method with the full spectrum of radiation or within a selected band from the high-energy part of the spectra makes it possible to detect embedded targets using their X-ray absorption properties with a certain degree of probability. However, due to the higher density of the fine muscovite, its absorption properties approach those of beryl; therefore, the brightness levels in the CT slices obtained in the low- and high-energy emission bands (Figure 17a,b) for beryl and fine muscovite are quite close. The image obtained after the dual-energy transformation of these slices shows the enhanced contrast of beryl against the mixture of muscovite, and fine beryl inclusions may also be identified with a higher contrast. It is worth noting that the solid piece of muscovite in the dual-energy image (Figure 17c) becomes closer in terms of brightness to the mixture of fine muscovite. Figure 17a–c should be considered in terms of contrast detection of beryl. Multispectral images may be required to separate objects with similar contrast properties more effectively. Respective studies are planned for the near future. The edge effects occurring in dense muscovite particles during dual-energy processing of CT slices are caused by a shift in the experiment in the process of obtaining the data sets for radiation with low and high energy quanta.

If monoenergetic collimated beams are used, dual-energy processing of X-ray images may be used to study thin films and coatings, namely to distinguish the contents of substances with similar chemical compositions based on their absorption properties. Computed tomography may be used to examine the profile of coatings. Notably, dual-energy CT is a non-destructive testing method. For comparison, the distribution profile of substances is currently tested via ion etching or chipping of samples, i.e., the techniques involving destruction of the coating or the sample.

Pulsed sources with semiconductor opening switches differ from traditional filament cathode X-ray sources in terms of their spectral distribution, which makes them more effective in dual-energy processing of X-ray images. The duration of the radiation flash for a pulsed X-ray source with a peak voltage of 330 kV is 10s of nanoseconds and the pulse repetition rate reaches 300 Hz, which allows us to obtain sharp images of moving objects at speeds up to 10s of meters per second [26]. Linear detectors and filament cathode sources are currently used to detect objects on conveyors, and the motion speed is limited to 0.1–0.3 m/s.

## 5. Conclusions

This article provides calculations and experimental studies on the contrast-enhanced detection of beryl hidden in muscovite host rock using dual-energy processing of X-ray images. The calculations are unique as they are performed for actual bremsstrahlung spectra, taking into account conversion of X-ray radiation into the detector output. The calculations show that detecting a 1–5 mm inclusion of beryl in a solid piece of muscovite of over 10 mm requires a combination of effective radiation energies: low and high energy of approximately 65 keV and 180 keV, respectively. These data were confirmed via a series of experiments using pulsed X-ray sources. These sources use semiconductor opening switches, rather than gas-filled spark gaps, to generate a voltage pulse that makes it possible to operate at a high pulse repetition rate and, as a consequence, with a higher sample motion speed. A series of experiments showed that a source with the voltage of 330 kV is the most suitable option for detecting beryl in muscovite with a total thickness of up to 70 mm.

Pulsed X-ray sources have short X-ray pulse duration and high pulse power, which allows us to use flat panel detectors with a pixel size of 0.1–0.2 mm on conveyor belts, rather than X-ray linear detectors with a pixel size of about 0.8 mm. Successful experiments were conducted using the VIVIX-V 2323D flat panel detector within this research. With an X-ray pulse duration of about 50 ns, blurring of moving objects in the X-ray image did not occur. This allows us to improve resolution, and the larger detector area makes it possible to increase the object speed on the conveyor belt up to several m/s. Continuous radiation sources have low currents and can only be used with linear detectors via a relatively large area on conveyors, which results in low resolution and acceleration to only 0.1–0.2 m/s.

The method of separating low and high energy using radiation filters rather than voltage control has been proven effective in this study. This result is due to the fact that the emission spectra of pulsed radiation sources contains a large proportion of low-energy quanta. The calculations also showed that the function of converting X-ray radiation into a VIVIX-V 2323D detector output positively affects the results of dual-energy image processing, when the radiation is not monoenergetic. In this case, the low-energy band of radiation was obtained using a 3 mm aluminum filter and calculation algorithm to remove high-energy quanta, and the high-energy band of radiation was obtained using a 3.5 mm copper filter. In the experiments, 5 mm thick beryl hidden in muscovite with the total thickness of 70 mm was effectively detected. Calculations and methodology for detecting hidden elements have independent value, as they allow us to investigate rocks with a thickness of over 50 mm, and can be applied to any X-ray spectra for dual-energy image processing.

Additionally, a pulsed source with a SOS has an advantage for CT. It is synchronized with the detector, and generation of X-ray pulses happens at the moment when the detector is registering the signal (“X-ray exposure”); when the detector transmits the signal to the computer (“read out”), X-ray radiation is not generated, unlike with constant current sources. Using pulsed X-ray sources with a SOS can improve image quality in spiral CT and lead to reduced radiation doses.

The results of this study and the methods of dual-energy processing of X-ray images may be further applied to other substances and minerals, when the task is to enhance the contrast in X-ray projections and CT images.

## Figures and Tables

**Figure 1 sensors-23-04393-f001:**
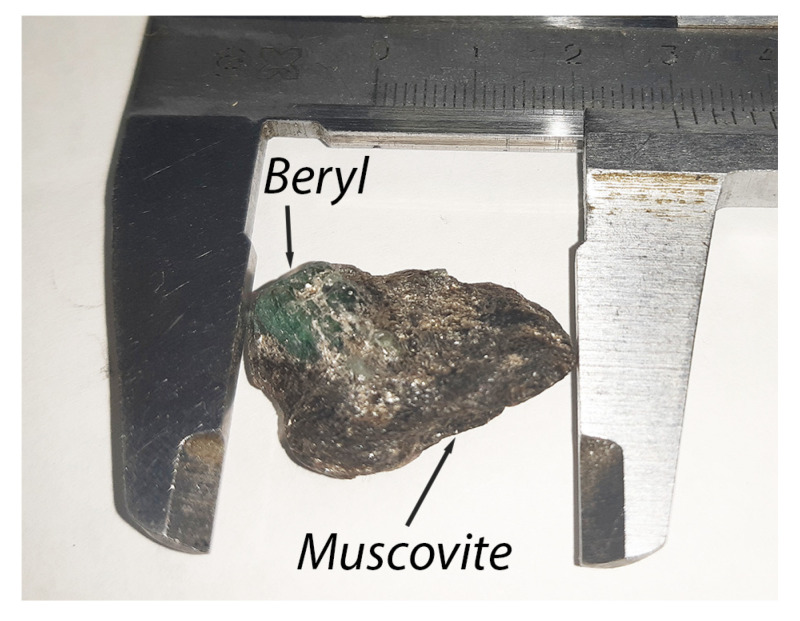
Sample of mineral beryl inside rock muscovite.

**Figure 2 sensors-23-04393-f002:**
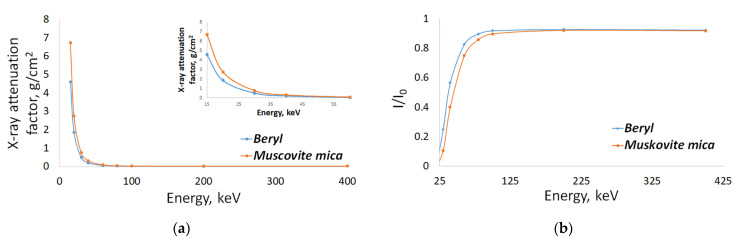
Attenuation of X-ray radiation in beryl and muscovite from radiation energy: (**a**) X-ray attenuation factor; (**b**) Attenuation of X-ray radiation due to absorption for an object 10 mm thick.

**Figure 3 sensors-23-04393-f003:**
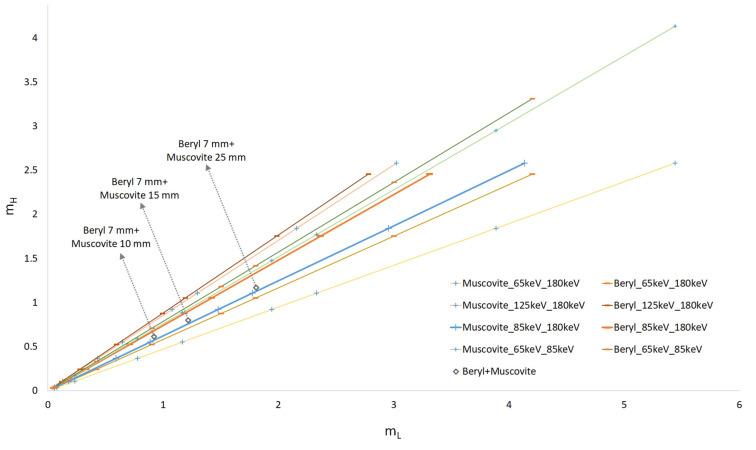
Representation of substances into axis (*m_L_*, *m_H_*) for different pairs of monoenergies.

**Figure 4 sensors-23-04393-f004:**
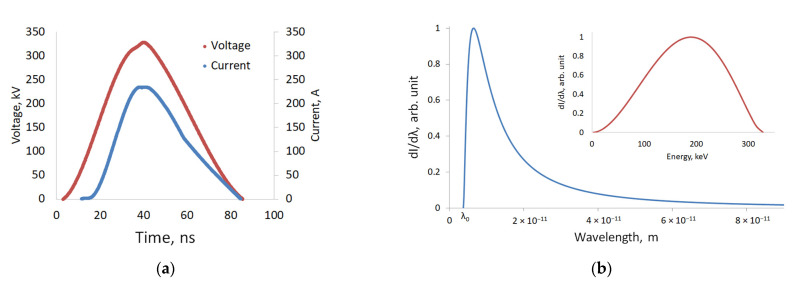
Characteristics of a pulsed X-ray source with voltage of 330 kV: (**a**) Voltage and current pulses; (**b**) emission spectra.

**Figure 5 sensors-23-04393-f005:**
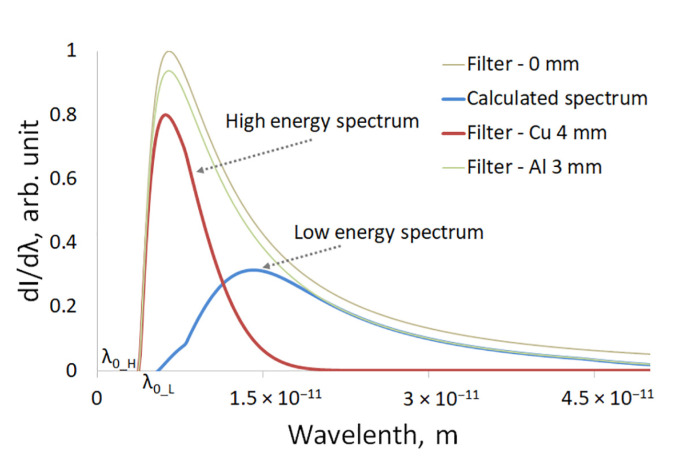
Spectra after filter and after applying the mathematical algorithm.

**Figure 6 sensors-23-04393-f006:**
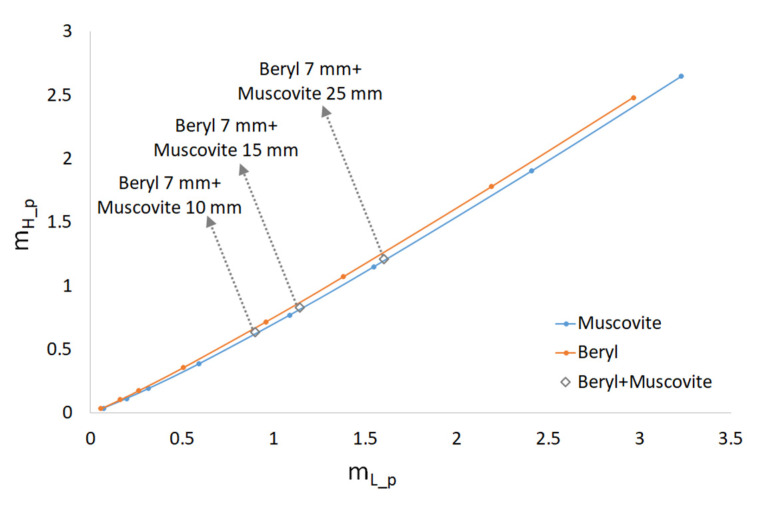
Representation of substances into axis (*m_L_*, *m_H_*) for real spectra of a pulsed X-ray source.

**Figure 7 sensors-23-04393-f007:**
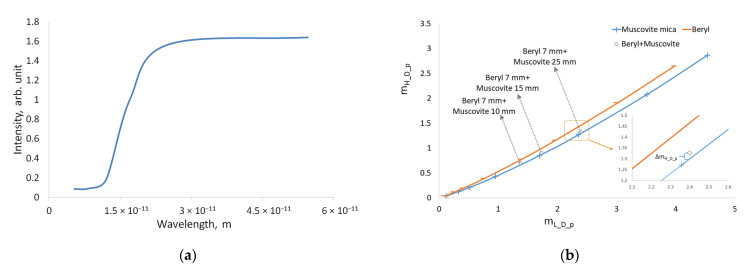
Influence of CsI scintillator: (**a**) Scintillator light emission efficiency *D*(*λ*) depends on the wavelength of the radiation; (**b**) Representation of substances into axis set composed (*m_L_*, *m_H_*) of real spectra of a pulsed X-ray source using function *D*(*λ*).

**Figure 8 sensors-23-04393-f008:**
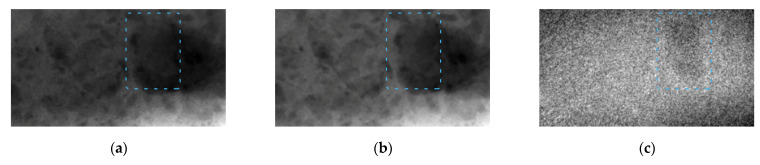
Images of beryl in muscovite taken from a pulsed X-ray source with peak voltage of 91 kV; beryl is indicated by blue dashed line. (**a**) Image obtained through transmitting low-energy radiation (47 keV) through sample; (**b**) image obtained through transmitting high-energy radiation (65 keV) through sample; (**c**) image obtained after dual-energy processing of these images.

**Figure 9 sensors-23-04393-f009:**
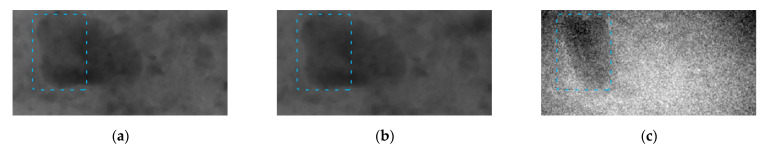
Images of beryl in muscovite taken from a pulsed X-ray source with the peak voltage of 330 kV; beryl is indicated by blue dashed line. (**a**) Image obtained through transmitting low-energy radiation (65 keV) through sample; (**b**) image obtained through transmitting high-energy radiation (180 keV) through sample; (**c**) image obtained after dual-energy processing of these images.

**Figure 10 sensors-23-04393-f010:**
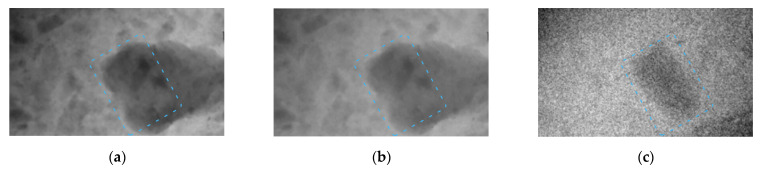
Images of beryl in muscovite taken from a pulsed X-ray source with peak voltage of 600 kV; beryl is indicated by blue dashed line. (**a**) Image obtained through transmitting low-energy radiation (100 keV) through sample; (**b**) image obtained through transmitting high-energy radiation (210 keV) through sample; (**c**) image obtained after dual-energy processing of these images.

**Figure 11 sensors-23-04393-f011:**
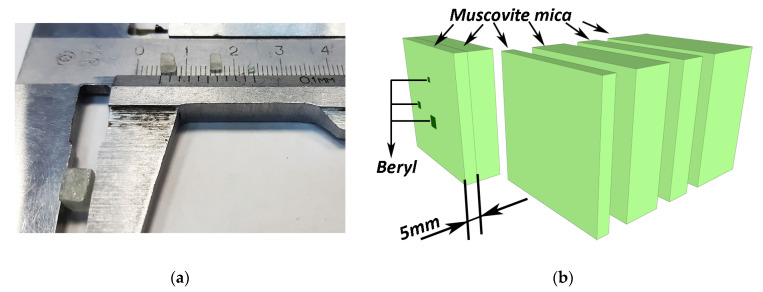
Scheme of experiment for detection of beryl particles inside muscovite: (**a**) size and shape of beryl particles; (**b**) Muscovite plates of varying thicknesses, with total thickness up to 70 mm, while one of the plates contains particles of beryl.

**Figure 12 sensors-23-04393-f012:**
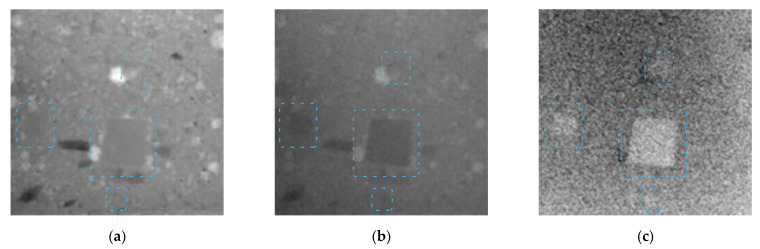
For X-ray images of a 5 mm thick muscovite plate with beryl particles, a pulsed X-ray source with the voltage of 330 kV is used; beryl is indicated by the blue dashed line. (**a**) Image obtained through transmitting low-energy radiation (65 keV) through sample; (**b**) image obtained through transmitting high-energy radiation (180 keV) through sample; (**c**) image obtained after dual-energy processing of these images (applied inversion).

**Figure 13 sensors-23-04393-f013:**
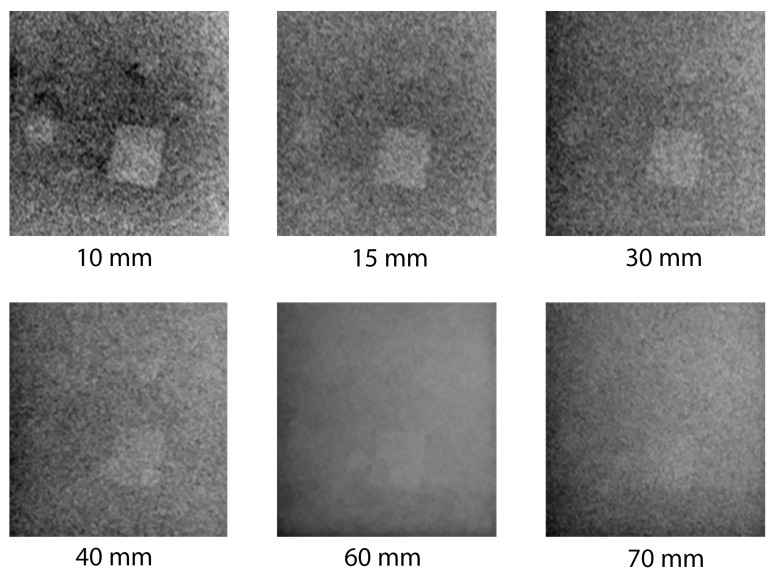
Result of dual-energy image processing for embedded beryl particles in muscovite mica up to 70 mm thick (applied inversion).

**Figure 14 sensors-23-04393-f014:**
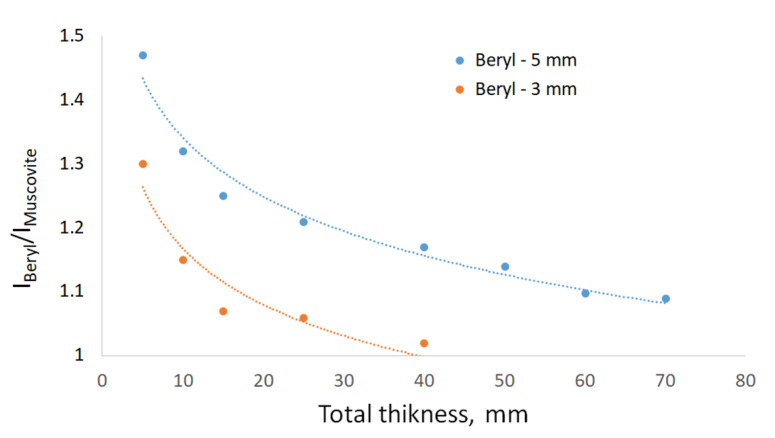
Visibility of beryl particles in muscovite after dual-energy X-ray image processing.

**Figure 15 sensors-23-04393-f015:**
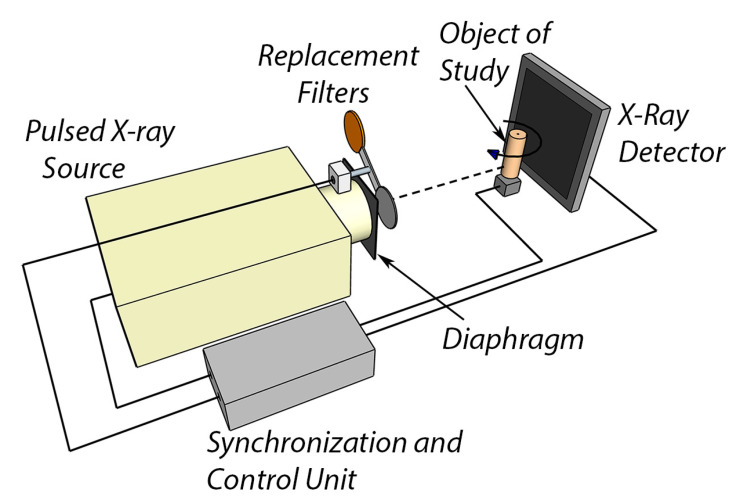
Scheme of experiment with computed tomography.

**Figure 16 sensors-23-04393-f016:**
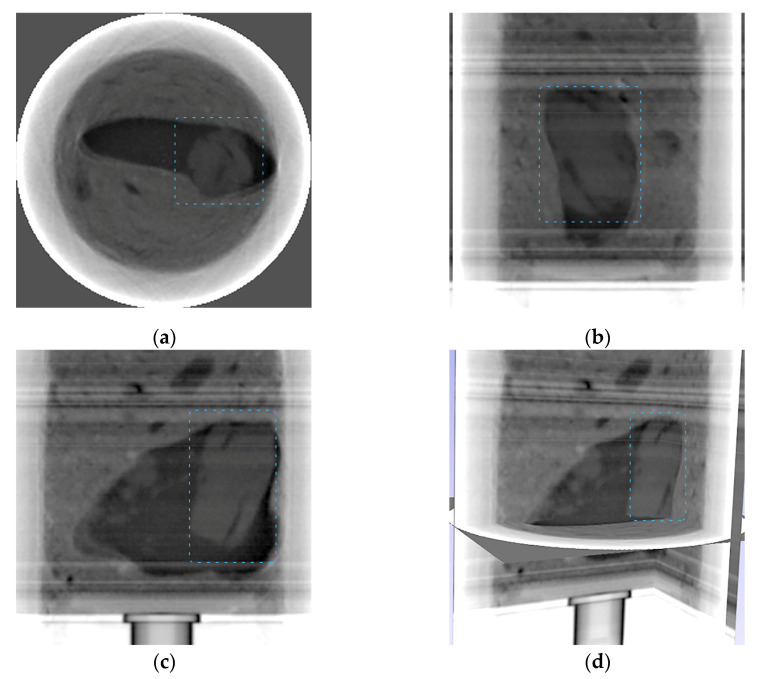
Depiction of 3D rendering of an experiment with computed tomography; beryl is indicated by the blue dashed line. (**a**–**c**) Resulting slices for the low-energy radiation in orthogonal projections; (**d**) 3D view.

**Figure 17 sensors-23-04393-f017:**
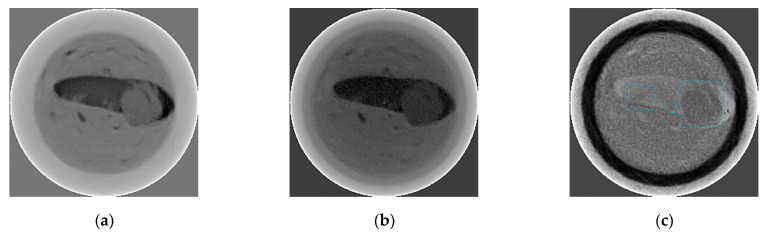
Dual-energy processing of CT slices: (**a**) CT scan taken at low energy (65 keV); (**b**) CT scan taken at high energy (180 keV); (**c**) image obtained after dual-energy processing of these images. Beryl is indicated by blue dashed line.

## Data Availability

Not applicable.

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
