# Peer review of "Dual-Energy Processing of X-ray Images of Beryl in Muscovite Obtained Using Pulsed X-ray Sources"

_sensors, 2023, doi:10.3390/s23094393_

Round 1
Reviewer 1 Report
This manuscript presents the technique of dual-energy imaging application for mineral rocks analysis. In my opinion the study is interesting and relevant. However, the following significant remarks can be done according to the text of the manuscript.
- Figure 4b. The Figure presents bremsstrahlung spectrum for the pulsed X-ray source of 330 kV. How this spectrum was calculated? How authors consider parameters of electron beam of the pulse source? Please clarify. Did this dependence plotted for monoenergetic electron beam without divergence or more precise model was used?
- Figure 4b. As I understood, only bremsstrahlung part of X-ray spectra are considered in further calculations and data processing. Real X-ray spectra include intensive characteristic lines. Did authors consider how characteristic X-ray radiation affects their calculations?
- Figure 5. What exactly means “low energy” spectrum in the figure? What is physical meaning of this spectrum? Is it kind of spectrum which somehow corresponds low-energy images which further got by applying authors algorithm for image processing? How exactly high and low energy parts separated?
- Figures 8-10. Figures presents images processed via special authors algorithm for separating low and high energy parts. I would recommend describes in details authors algorithm. I believe it is the most interesting part of the study, but currently the manuscript is suffering from a lack of information about this algorithm.
- Figure 17. Please, prove somehow that the quality of CT images is higher for dual energy technique. From the figure for me it seems that the beryl is indicated easily for all three images, while dual energy image is more “noisy”.
In conclusion, I would say that the dual-energy technique is important in imaging, however, currently more advanced approaches which allows considering few spectral regions are widely used for imaging (“color” or “spectral” imaging). These new approaches became especially popular with the advent of pix-detectors and the algorithm for obtaining few “spectral” images in one shot with this kind of detectors are known (e.g. Yokhana, V. S., Arhatari, B. D., & Abbey, B. (2022). Materials Separation via the Matrix Method Employing Energy-Discriminating X-ray Detection. Applied Sciences, 12(6), 3198.). I would recommend to consider these techniques in the review. Another important issue is X-ray energy optimization, but this question is much more complex than simple selection of three different sources. I understand perfectly, that author used sources which they have, but the study would benefit from a few words about modern approaches for X-ray energy optimization. In this regard, I would recommend study [Gogolev, A. S., et al. "Tomography Imaging Taking Into Account Spectral Information." Bulletin of the Lebedev Physics Institute 45.6 (2018): 176-181.]. I would also suggest to authors considering for review another important application besides medical application and rock analysis, where dual energy technique is widely use - Multiphase flow metering (e.g. Gogolev, A. S., Cherepennikov, Y. M., Vukolov, A. V., Rezaev, R. O., Stuchebrov, S. G., Hampai, D., ... & Polese, С. (2015). WD-XRA technique in multiphase flow measuring. Nuclear Instruments and Methods in Physics Research Section B: Beam Interactions with Materials and Atoms, 355, 276-280.). Finally, I would recommend proofreading the manuscript’s text by a native speaker.
Nevertheless, I believe that the article can be recommended for publication with the corrections indicated.
Author Response
We thank the reviewer for carefully reviewing our article. We agree with the comments provided and have made appropriate corrections to the article as follows:
1-2. Changes made in the article, see lines 259-264.
- Change made in the article, see lines 399-400.
- Change made in the article, see lines 373-379. We also modified Figure 7 (b) for better understanding of the algorithm.
- We added a comment in the discussion section, see lines 567-574.
In the introduction, we added the scientific works suggested by the reviewer, which relate to the topic of our research. We also rephrased some sentences for clarity.
Reviewer 2 Report
The authors used a pulsed X-ray source for dual-energy X-ray imaging. It is well known that use of two energies enhances the density resolving power in X-ray imaging. A detailed explanation on the analysis of the images that were acquired with a white X-ray beam is given. However, the technique is already too established to make this work meaningful. Since similar experiments were made in the early days in development of dual-energy CT, the theories and results are not novel. One notable feature of this paper is the use of a pulsed X-ray source, but the advantages of this X-ray source such the high repetition rate are not used at all. From these reasons, I cannot recommend publication of this paper.
There are other issues as well:
The authors used metal filters to modify the X-ray spectrum. This is a rather primitive method. Nowadays, dual-energy X-ray imaging, particularly CT, is widely used in hospitals. In this case, X-ray energy is changed by using different voltages in the X-ray generator(s). Since filtering shifts the X-ray spectrum towards the higher energies, combination with a voltage control that shifts the high end of the spectrum is more efficient. Although data obtained at different source energies are shown in this paper, more studies are necessary to find the best combinations.
Identification of beryl in muscovite may have an industrial application. The technique needs much sophistication because the rude ores are very variable in sizes and compositions. It is uncertain if this pulsed X-ray source is the best for the purpose.
One interesting idea is using detector sensitivity as a method to detect low energy X-rays only. However, this is usually avoided because X-rays that pass through the scintillator create high background and also damage the sensor.
The manuscript is generally written well with a few exceptions.
Line 233: The “mathematical algorithms” are probably a method to calculate difference between images obtained with different X-ray spectra. This should be explained more clearly.
After Figure 4, X-ray spectra are shown in wavelength, making it hard to compare them with those in Figure 2 that are shown in energy.
Author Response
We appreciate your feedback and comments.
The dual-energy method is indeed widely used, but mainly applicable for separating objects that differ greatly in X-ray absorption properties (bone structure and muscle tissue, metal and plastic). This is not the case for beryl in muscovite.
The pulsed X-ray sources with semiconductor opening switches that we develop in our laboratory and use in this study have great potential for imaging moving objects. We conducted such experiments and provided a reference in the article (26). We added a comment about this in the article, see lines 587-592.
We understand that ore usually has differences, depending on the deposit and many other factors. However, calibrations can be made for specific deposits. Usually, when searching for valuable minerals, the ore is divided into fractions. For a certain fractional composition, the probability of finding large beryls will be high. These are technical issues and questions of economic feasibility. We wanted to show in the article the fundamental possibility of using the dual-energy method for substances with similar X-ray absorption properties, mainly for 2D projection. If you are interested, we are able to localize beryls with high probability on dual-energy images, both through the use of artificial neural networks and standard methods for object localization.
The X-ray panel we used can be used for radiation with an energy of 450 keV, as stated by the manufacturer. Of course, its lifetime will be shorter than when used for radiation with an energy of 60-100 keV (perhaps the panel will work not for 4-5 years, but for 2-3 years). At high radiation energies, electronics suffer primarily, as far as we know from communicating with dealers of flat-panel detectors. However, these are questions of economic feasibility.
More details about the calculation algorithm have been added in lines 373-379 and in Figure 7(b).
Figure 4(b) has been modified, and a spectrum has been added where energy is plotted on the abscissa axis.
Reviewer 3 Report
The authors report calculations and experimental verification as a method for dual-energy processing of 2D and 3D x-ray images collected using a pulsed x-ray source for the detection of beryl in muscovite mica via contrast enhancement by utilizing mentioned dual processing. In their experiments that employed different maximum voltages in pulsed x-ray imaging experiments, the authors managed to find conditions where the two substances having very similar x-ray absorption characteristics (and thus similar grayscale intensities) were differentiated in the images. They also report on modeling of the emission spectra and selecting the necessary energy bands where dual-energy image processing would result in appropriate image contrast between beryl and muscovite. They found that the pulse voltage of 330 kV was the best condition to detect beryl with the content of 5–7% against muscovite background.
The study is well-planned and executed, the results clearly explained. The best part of the report to me is that the experimental conditions were also supported by calculations and that a source available at their laboratory could be used to achieve very impressive contrast on a small amount of beryl content.
I have the following comments and requests regarding the manuscript, which requires some revision:
- A few more references should be found and included. In line 36, the authors mention: “Some research papers suggest the use of X-ray absorption methods in exploration for valuable minerals.” with no reference cited. The subsequent citations do not really address this issue, so please add some citations to this statement.
- In line 177, it is written that monoenergetic radiation sources are not used in practical applications due to their low radiation intensity (low attenuation). Instead, sources with a bremsstrahlung spectrum are used. I want to note that while this is largely true for laboratory x-ray sources, but not exactly true for synchrotron sources where the intensity can be much greater even for soft, monochromatic x-rays. Please include this in your discussion.
- Line 432: It is stated that: “The best contrast-enhanced image in detection of beryl against a background of muscovite is obtained in the ideal case with monoenergetic radiation. A pulsed X-ray source with a bremsstrahlung spectrum was used in the experiments…” While this reviewer understands what the authors meant, I think this may be a bit confusing to some readers, since it sounds like the mono radiation is what the authors prefer, but they still opt for polychromatic radiation. Please rephrase to avoid this possible confusion.
Overall, interesting work with good potential applications.
Author Response
We appreciate the reviewer finding our work interesting and not wasting their time reading our manuscript. We agree with the suggested revisions and have made the appropriate changes in the article:
References 3, 4 have been added to the article.
Line 177 has been modified, please see lines 214-218.
Line 432 has been modified, please see lines 505-507.
Reviewer 4 Report
The paper presents a method for dual-energy processing of X-ray images using pulsed X-ray sources for the contrast detection of beryl in muscovite mica in 2D X-ray and CT images. A pulsed X-ray source with the pulse voltage of 330 kV was chosen for the contrast detection of beryl with the content of 5–7% against the background of muscovite with the thickness up to 70 mm. The research work has certain innovation and application value.
Author Response
We are glad that you found the research interesting.
Round 2
Reviewer 1 Report
I thank the authors for the corrections made to the manuscript. I believe the manuscript will benefit from these corrections. However, I would recommend that the authors consider discussed points for future publications. Besides, I still recommend professional proofreading of the manuscript for language corrections.
Author Response
Thank you for your comments, you have helped to improve the quality of the work. Initially, we had the translation done by a specialized company, where scientific and technical specialists carried out the translation. They also had the translation checked by an editor.
We took your advice and consulted another specialist (a native speaker). He conducted an extensive proofreading of the text and made significant corrections. We hope that this has improved the quality of the article.
Reviewer 2 Report
It is hard to say this system is more suitable for identification of beryl in muscovite than other available systems because the advantages of the X-ray source is not utilized. Using an apparatus simply because it is available is not a good reason to choose it in applications.
Author Response
Thank you for wanting to help improve our work. We have made changes to the conclusion section.
We would like to clarify that, in our opinion, there are advantages to using pulsed radiation sources. Firstly, it increases the resolution capability, as flat panel detectors with smaller pixel sizes can be used on conveyors instead of linear detectors. Secondly, it may increase conveyor speed, as the X-ray pulse duration is short. Thirdly, due to the voltage rise and fall fronts, the spectrum of the pulsed source differs from that of a constant source, which is also a positive characteristic when extracting a certain energy band from the spectrum. We believe that these are unique features that are of interest to potential readers of the article and have practical application prospects.
There are also advantages to using pulsed sources for CT, which we briefly described.
We also demonstrated the ability to detect useful minerals hidden in large thicknesses of barren rock, over 50 mm. The X-ray absorption properties of the mineral and rock are similar, and there are no similar studies in the world. An interesting effect was obtained when taking into account the detector function when using dual-energy calculations. The calculation methodology differs among other researchers.
From our perspective, this is a comprehensive study that can be useful for many scientific and technical specialists.